# BEV-CLIP: Multi-modal BEV Retrieval Methodology for Complex Scene in Autonomous Driving

## Abstract

The demand for the retrieval of complex scene data in autonomous driving is increasing, especially as passenger vehicles have been equipped with the ability to navigate urban settings, with the imperative to address long-tail scenarios. Meanwhile, under the pre-existing two dimensional image retrieval method, some problems may arise with scene retrieval, such as lack of global feature representation and sub-par text retrieval ability. To address these issues, we have proposed **BEV-CLIP**, the first multimodal BEV retrieval methodology that utilize descriptive text as an input to retrieve corresponding scenes. This methodology applies the semantic feature extraction abilities of a large language model (LLM) to facilitate zero-shot retrieval of extensive text descriptions, and incorporates semi-structured information from a knowledge graph to improve the semantic richness and variety of the language embedding. Our experiments result in 87.66% accuracy on NuScenes dataset in text-to-BEV feature retrieval. The demonstrated cases in our paper support that our retrieval method is also indicated to be effective in identifying certain long-tail corner scenes.

## 1 Introduction

In recent years, a growing focus has been raised on the retrieval task in the field of autonomous driving. A well-designed retrieval method is essential for addressing corner cases in autonomous driving data (Bogdoll et al., 2021). However, corner case scenarios often contain instances or features that rarely occurs. For example, unprotected left turn scenario describes ego-vehicle is turning left without the protection of a left turn traffic light. In scenarios such as this, all necessary road participants (e.g opposite vehicle, lane line and vehicle in the left neighbouring lane) are distributed in a unique pattern requiring global abstraction. Meanwhile, the precise description text of road participants may be extremely customized in specific cases, and is unable to be included in pre-existing labels from any dataset. Hence, the retrieval model desires the capability to represent complex features that are distributed over a wide range of scenes (Li et al., 2022a).

This paper aims to study two fundamental problems towards developing a system for image-text retrieval in autonomous driving scenes. **(1)** *How can we overcome the limitations intrinsic to two-dimensional image features, particularly their poor capability to effectively represent global feature within autonomous driving scenarios?* **(2)** *Which methodologies could potentially enhance the currently unsatisfactory efficacy of text representations within the field of autonomous driving?* To address these two issues, we suggest the following.

**Feature extraction**  We suggest the utilisation of the Bird's-Eye View (BEV) framework as it offers a unified representation for autonomous driving scene description. By combining multiview camera data, the BEV framework converts 2D perception into a detailed 3D description from a top-down perspective (Xie et al., 2022; Huang et al., 2021; Li et al., 2023b). This approach overcomes limitations associated with feature truncation, which frequently occurs in single-view approaches, and enables better downstream tasks. As a notable solution, BEVFormer (Li et al., 2022b), a transformer-based BEV encoder, generates global features from camera input alone and serve as an end-to-end model for various downstream tasks. Thus, performing scene retrieval on BEV features is a integrated solution to address the problem of extracting global representation, and as a

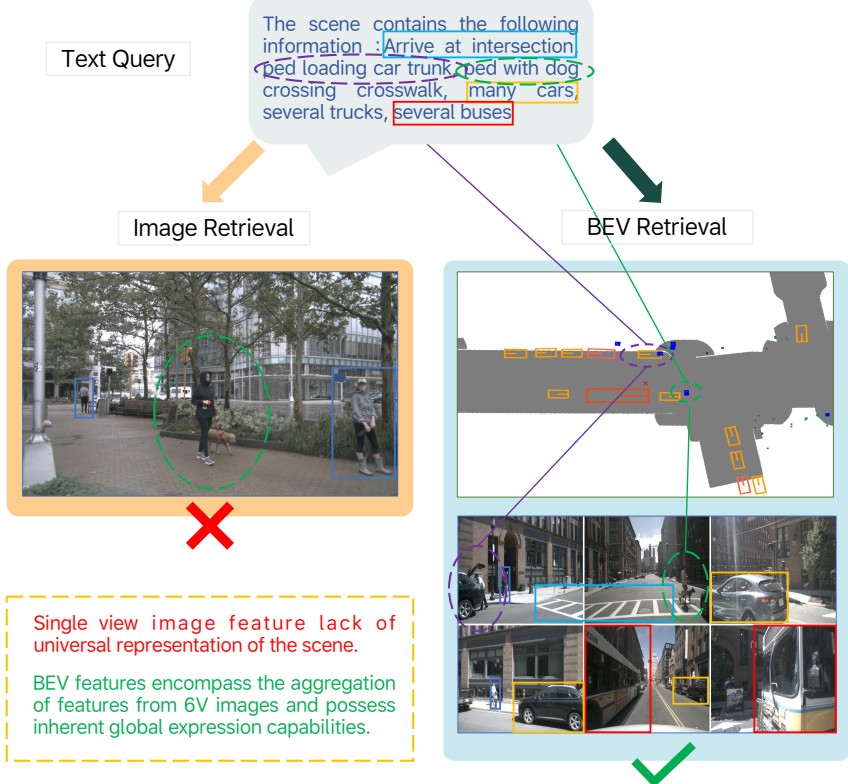

Figure 1: BEV-CLIP, the first BEV retrieval method retrieves corner cases on autonomous driving. In contrast to 2D image retrieval, BEV-CLIP allows semantic retrieval related to complex global features in the context of BEV features. Meanwhile, BEV-CLIP uses a Large Language Model (LLM) to enhance the model's ability to understand complex descriptions in the retrieved text.

well-known method, incorporating BEVFormer for BEV feature extraction is both advantageous and justified.

**Language representation** We suggest the incorporation of intricate semantic data as an additional input to compensate for abstracted features not evident solely in image data. Existing multimodal large language models (LLM) have demonstrated remarkable capabilities in expressing features of other modalities (Huang et al., 2023; Liu et al., 2023). CLIP (Radford et al., 2021) presents a baseline for multimodal retrieval, enabling the model to generate zero-shot inferences by leveraging the language model's decoding capabilities. Inspired by this, we construct an improved LLM with fine-tuning strategies to provide more richness semantic information as supplement for BEV feature. Additionally, knowledge graph features will be incorporated to enhance the salience of knowledge in the autonomous driving domain. The fusion of the LLM and knowledge graph aims to achieve excellent cross-modal understanding in our method.

In this paper, we propose BEV-CLIP, the first BEV retrieval method. It unifies capabilities of BEV feature aggregation and representation, and rich semantic abstraction abilities from LLM and knowledge graph. Our key designs can be summarized as follows: (a) A novel method to perform BEV retrieval and BEV caption generation. (b) A well-performed assembly method of LLM and knowledge graph to improve the generalisation ability of language comprehension. (c) An efficient structure called shared multimodal prompt (SCP) that bridges the gap between BEV and language branch to provide a well-fused feature representation before contrastive learning.

Our contributions can be summarised as follows:

1. We propose a retrieval method based on BEV feature which can retrieve global features in autonomous driving scenarios and have a significant understanding capability of complex scenes. To the best of our knowledge, this is the first BEV retrieval method in the field of autonomous driving.

2. We propose a multimodal retrieval method powered by LLM and knowledge graph to achieve contrastive learning between text description and BEV feature retrieval for autonomous driving, so that it can perform zero-shot retrieval using long text descriptions.

3. We build a retrieval validation pipeline based on existing datasets and achieve a result at rank-1 of 87.66 % on the NuScenes dataset, fully verifying the effectiveness of optimising the BEV retrieval model.

## 2 RELATED WORK

**BEV Feature Acquisition**   Vision-dependent Bird's Eye View (BEV) perception has gained significant attention as it offers advantages in rendering complex scenes and facilitating the fusion of multiple camera inputs. (Garnett et al., 2019; Can et al., 2021) (Chen et al., 2022) proposed an approach based on Inverse Perspective Mapping (IPM), which inversely maps features in perspective space to BEV space. However, IPM is restricted to the ideal assumption of flat ground. (Philion & Fidler, 2020; Hu et al., 2022) proposed a method based on monocular depth estimation (MDE). However, the lack of explicitly supervised images prevents the effectiveness of MDE, which, in turn, affects the accuracy of BEV features.

In recent years, transformer architectures have been widely adopted for BEV models. Transformer uses a global attention mechanism, where the mapping of any position in the target domain to the source domain has the same distance, overcoming the limitation of CNN. (Wang et al., 2022; Can et al., 2021) converts the query to a 2D feature by projection, so that the network can find the real 3D obstacle features automatically. BEVFormer (Li et al., 2022b) combines spatial and temporal attention, creating a novel aggregation technique for BEV features. By iteratively updating the history frames with query information using an RNN-like method and subsequently passing it to the current frame, the computational burden is effectively managed. We claim that the inclusion of temporal information in BEV features is highly suitable for retrieval tasks, as it excels in reconstructing dynamic scenarios for autonomous driving.

**Retrieval tasks**   The field of cross-modal retrieval, which aims to bridge the representation gap between different modalities, has gained significant attention in the literature. One prominent approach proposed by (Radford et al., 2021) involves training a migratable visual model using text as a supervision signal. In this approach, both the text and image inputs are separately encoded by a text encoder and an image encoder, respectively, generating corresponding feature representations that can be utilized for contrastive learning. Through extensive training with a large amount of data, the two encoders achieve good generalization capabilities, enabling zero-shot retrieval. Based on this, (Li et al., 2023a) introduces a multimodal Encoder-Decoder structure, which effectively perform multitask pre-learning and transfer learning. Additionally, it introduces a learnable Q-Former structure for bridging the representation gap between modalities, so that the model can be fine-tuned using an LLM with frozen parameters while updating only a small number of parameters. (Khattak et al., 2023; Chen et al., 2023) proposed a joint prompt method that adds learnable context tokens to the main branch as implicit prompts to establish interaction between the image and text branches. We argue that this joint prompt method is able to perform supervised training for retrieval tasks based on large language models.

**Language Pretraining**   Recent research has demonstrated that the emergent ability of LLMs allows them to achieve a great improvement in comprehension after reaching a certain magnitude of token count. (Brown et al., 2020) demonstrated for the first time the superiority of autoregressive language modeling by showing that impressive performance can be achieved with few-shot/zero-shot inferencing through prompting and in-context learning. Moreover, several studies (Chowdhery et al., 2022; Touvron et al., 2023; Chung et al., 2022; Driess et al., 2023) have confirmed the efficacy of LLMs on modest tasks, leveraging a limited number of fine-tuned parameters across a diverse mixture of multi-task datasets. These findings collectively emphasize the significant potential of LLMs in enhancing various language-related tasks.

**Retrieval tasks based on knowledge graph**   Knowledge graphs are known for their exceptional scalability in handling unstructured data types. We explore the use of knowledge graphs in the

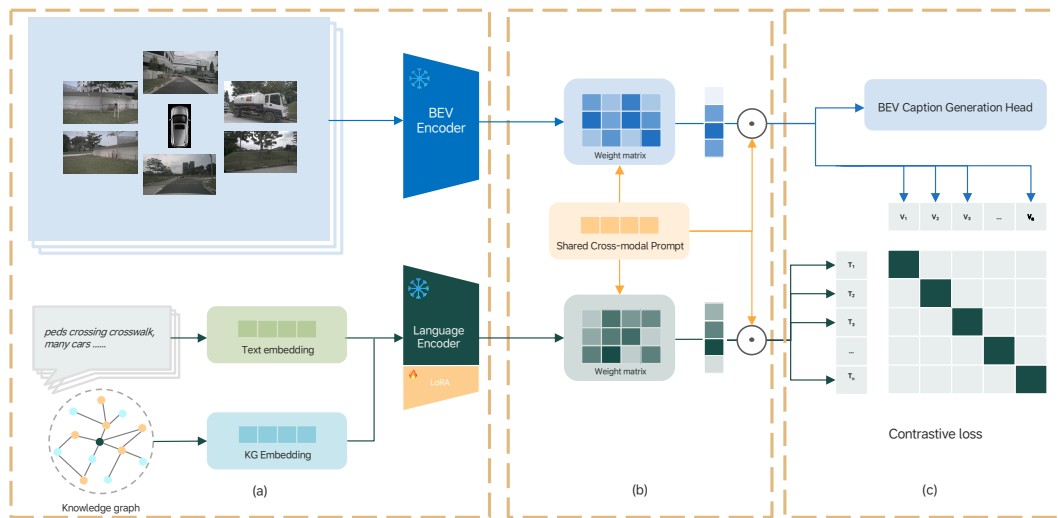

Figure 2: **Overall structure of BEV-CLIP.** (a) Processing of BEV and text features. The image from 6 surrounding cameras are generated into a BEV feature by the BEV Encoder with frozen parameters. At the same time, the input text embedding is concatenated with the keyword-matched Knowledge Graph node embedding, and fed into the Language Encoder with LoRA branch for processing. (b) Shared cross-modal prompt (SCP), which aligns the BEV and linguistic features in the same hidden space. (c)Joint supervision of caption generation and retrieval tasks. ⊙ denotes dot product.

Retrieval-augmented Generation (RAG) domain and draw inspiration from it to complement generative LLMs. There are several studies in the existing literature that combine multiple kinds of knowledge to augment language models, such as augmenting common-sense reasoning with knowledge graphs (Yu et al., 2022) and introducing multimodal visual features to augment affective dialog (Liang et al., 2022).

## 3 METHOD

In this section, we discuss main structure of BEV-CLIP, a methodology of text-to-BEV contrastive learning retrieval. We obtain the method of utilizing the pre-trained BEV encoder weight for retrieval tasks and apply cross-modal interacting strategy with the language representation, which fused by text description and knowledge graph embedding.

### 3.1 GLOBAL BEV FEATURE

In our approach, we have explored multiple methods of acquiring BEV feature, and realized that visual based BEV approaches have the most universality for autonomous driving applications. Therefore, we adopt BEVFormer as the baseline model for BEV feature extraction. BEVFormer is a dedicated camera-based BEV perception model that incorporates two critical modules: spatial attention and temporal attention. These modules enable the aggregation of spatial and temporal information, facilitating the characterisation of moveable obstacles from multiple perspectives. Thus, the generated BEV features possess a higher capacity to encapsulate the entire scene. It is worth mentioning that our approach can accommodate various feature extraction networks that integrate multi-view images or point cloud data into BEV features.

In the retrieval task, we keep all parameters of the BEVFormer model frozen and employ the generated features directly for downstream post-processing and retrieval. Consequently, the approach minimises the overheads associated with training and enhances the overall efficiency of the retrieval method.

### 3.2 KNOWLEDGE GRAPH PROMPTING

In the context of autonomous driving scene description, semantic information related to the scene often exhibits discrete characteristics. To address this, we propose the incorporation of unstructured information that complements the descriptive text by providing associative information. In

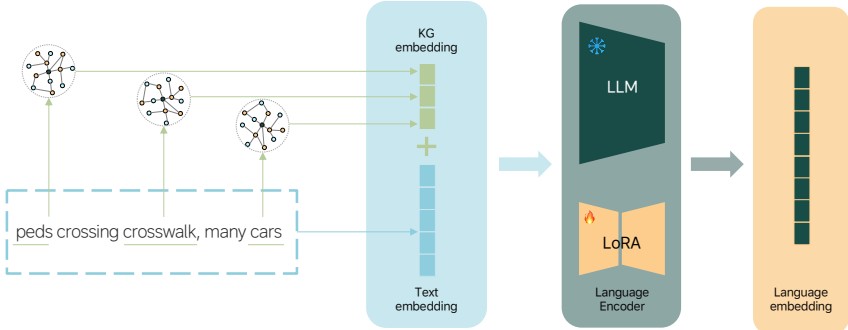

Figure 3: **Knowledge graph keyword matching.** We generate the knowledge graph embedding of autonomous driving domain by pre-training, and concatenate the embedding that has appeared in the input text with the tokenised text in order, and jointly input them into the language encoder. The language encoder is a structure with frozen LLM parameters and a LoRA branch for fine-tuning.

our approach, we leverage a Graph Neural Network (GNN) to train a knowledge graph in the field of autonomous driving. Each node in the graph corresponds to a keyword relevant to autonomous driving, and the embeddings associated with these nodes capture the associative representation of autonomous driving keywords. Subsequently, these keyword embeddings are concatenated into the tokenised sentence, thereby expanding the representation of the encoded text.

| Subject | Predicate | Object |
|---|---|---|
| inst:scene | rdf:type | Object:Scene |
| inst:scene | rdf:includeType | Object:Vehicle |
| inst:scene | rdf:include | inst:vehicle |
| inst:vehicle | rdf:participate_in | inst:driving |
| inst:driving | rdf:type | Movement:Driving |
| inst:vehicle | rdf:participate_in | inst:driving |
| inst:driving | rdf:has_participant | inst:vehicle |

Table 1: Example Resource Description Frameworks (RDF) in PandaSet

In order to acquire comprehensive knowledge in the domain of autonomous driving, it is essential to extract and generalize relationships within a knowledge graph across a wide range of instances. Constructing perceptual data into graph instances that can be effectively learnt enables this process. To achieve this, (Wickramarachchi et al., 2020) proposed a knowledge graph specifically tailored for the field of autonomous driving. This graph is constructed using scene-aware data obtained from PandaSet (Xiao et al., 2021), and abstracts triplets (as illustrated in Table 1) to establish associations between perceptual instances, labels, and actions. In accordance with our specific requirements, the entities within the knowledge graph are categorized into three main groups: instances, objects, and movements.

Within the knowledge graph, a predominant portion of nodes corresponds to instances, while object and movement nodes connect to numerous distinct instance nodes. Exploiting the extensive connectivity between object and movement nodes to extract there relations serves as our primary objective, enabling the acquisition of their associative representations. To achieve this, we employ an embedding technique in which entities and relations present in the knowledge graph are mapped into a continuous vector space. Through training, we iteratively optimise and regress suitable vectors within the vector space for each node. Drawing inspiration from the approach introduced by TransE (Bordes et al., 2013), based on translational distance modelling, we adopt a distance-based scoring function to assess relationships. For each triplet in the graph, the following scoring function is defined:

$$f_r(h,t) = - \parallel h + r - t \parallel_{norm} \tag{1}$$

Here, $h, r, t$ stands for subject, predicate, object, respectively. The $norm$ is generally defined as the $L1$ or $L2$ norm. Following the application of the aforementioned scoring functions, the embedding vectors associated with nodes within the graph successfully capture the translation transformation relations represented by all the triplets present.

To explore more strategies of acquiring more structured representation of relations within the knowledge graph, we also adapt DistMult (Yang et al., 2014) and ConvE (Dettmers et al., 2018) as alternative knowledge graph embeddings. DistMult is a bi-linear model that calculates the credibility of entities and relationships in vector space. ConvE is a model that uses two-dimensional convolution to achieve link prediction. ConvE first converts the head entity and relationship into two-dimensional vectors, then uses a convolution layer and a fully connected layer to obtain interaction information, then calculates with matrix W and tail entity to judge the credibility of the current triplet. We have tested these two approaches with our baseline method and found that DistMult performs well on symmetric relationships, while ConvE is advantageous with graphs containing high-degree nodes.

### 3.3 SEMANTIC REPRESENTATION FUSING

To achieve zero-shot retrieval, it is crucial to extract the comprehensive semantics information from text input. Pretrained models possess the potential to offer a substantial understanding of word knowledge, grammar, syntax, and overall semantics within general-purpose contexts. These models also exhibit proficiency in contextual comprehension and generation of coherent responses. Furthermore, in various end-to-end LLM approaches, the model demonstrates the ability to jointly characterize diverse modal information of serialization. According to the semantic understanding capabilities of existing LLMs, we fine-tune them for training within the domain of autonomous driving.

Leveraging sentence embedding, we integrate and merge graph embeddings to create a synthetic representation. Specifically, we index the keywords from the graph that appear in the text input and concatenate them into the text embedding sequence in the order of their occurrence. The resulting language embedding, fused with the graph representation, serves as the supervision for retrieval task and caption generation.

### 3.4 SHARED CROSS-MODAL PROMPT

In this section, we explain how the cross-modal interaction in our method is performed between the BEV and the text branch. In the early stage of our experiment, we have attempted to directly optimise the constrastive loss but it resulted in an unsatisfactory outcome. We realised that the components in both BEV and text branches are pre-trained on single-modal datasets. Meanwhile, most of the parameters of these components have to remain frozen to maintain an affordable training cost.

Inspired by Q-Former structure from BLIP2 (Li et al., 2023a), we design an independent structure to bridge the two modalities by implementing a cross-attention method, which is referred to as Shared Cross-Modal Prompt (SCP). These prompts comprise a set of long sequential tokens that are learnable under an autonomous feature space. Our intention is to leverage these prompts to map the BEV features and textual features onto the same manifold space, facilitating the alignment of the divergent modal information present in the two branches.

The learnable parameters in SCP can be represented as a sequence: $T = \{t_1, t_2, \ldots, t_k\}$, and the BEV features can be reshaped and compressed into a sequence of feature embeddings, $F = \{f_1, f_2, \ldots, f_n\}$. For each token, $t_i$, in the SCP sequence, the similarity of each feature $f_j$ in the sequence of BEV features to token $t_i$ can be computed as: $r_{ij} = sim(t_i, f_j)$, and the maximum value of its similarity can be obtained as the projection of the BEV feature $F$ on the token $t_i$: $r_i = \max_j(r_{ij})$. For all learnable tokens $T$, use the same method to obtain maximum similarities of $F$ on $T$ as a result of the projection of feature $F$ on $T$: $R = \{r_1, r_2, \ldots, r_k\}$. With the softmax function, $R$ can be converted into a weight for the sequence of SCP: $w_i = \frac{e^{r_i}}{\sum_j^C e^{r_j}}$. Such an operation is also performed in the same fashion for text branches. By assigning the weight to the SCP, the feature sequence obtained after the fusion of BEV features with SCP can be derived.

In our approach, SCP is shared between both the BEV and text branches, with the difference in output derived from distinct weight sequences assigned to the features within each branch. Through this fusion technique, it is ensured that the resulting features originate from the same embedding space and maintain an identical shape of the feature map.

To implement the contrastive learning task in CLIP, we substitute the image features with BEV features. This process involves using the outputs of the text branch and the BEV branch merged with the prompt as distinct sets of features, which are pooled to generate one-dimensional vectors. Subsequently, these vectors are employed for similarity calculations. Additionally, we introduce a caption generation task as an auxiliary component for model training. We employ the output of the BEV branch to generate captions. To fulfill this task, we utilize a lightweight decoder based on the Transformer structure, while the corresponding text description to the BEV sample serves as the supervision label for this task.

## 4 EXPERIMENT

### 4.1 DATASETS

**Nuscenes Dataset**   The NuScenes dataset (Caesar et al., 2020) is a large-scale public dataset for autonomous driving. The dataset contains a total of 1,000 driving scenarios collected in Boston and Singapore, with each scenario lasting about 20s. Each sample is captured by RGB cameras distributed in six different viewpoints of the car and a lidar placed on the roof. The training and validation sets contain a total of 34,149 key frames, and the dataset provides textual descriptions of each different scene and labelled 2D and 3D target detection results. We performed all the training and evaluation experiments on the NuScenes dataset, and in order to further improve the informative diversity of this textual description and to reduce the repetition of the caption, we briefly supplemented the original caption with the target detection results labelled in the scenes.

**Caption complement strategy**   For each keyframe sample, we obtain all the obstacle information in the current perception result through the object detection labelling result. Concurrently, the frequency of each obstacle type's occurrence is quantified. In order to mitigate the complexity of the training task, the quantity descriptors in the tabulated results are rendered more ambiguous, and supplement the caption text with *e.g.*, "many cars, several trucks, one bus, one bicycle, one motorcycle, several traffic cones". We concatenated the supplemented text directly after the original caption text as the scene description for our training and evaluation.

**Knowledge graph data**   We utilise autonomous driving knowledge graph (ADKG) proposed by (Wickramarachchi et al., 2020) as knowledge graph data to train knowledge graph embedding for keywords in the field of autonomous driving. ADKG is generated based on perceptual data from Hesai's PandaSet (Xiao et al., 2021). It contains more than 57,000 instances and more than 330,000 triplets. It also contains 7 action labels and more than 40 object labels. Based on these labels, we manually perform synonym mapping so that the caption mentioned above can trigger the knowledge graph retrieval more frequently and accurately.

### 4.2 IMPLEMENTATION DETAILS

In the following experiment, the default size of our BEV feature map is (2500,1,256), the hidden size of the LLM embedding is set to 4096. To control the experimental variables, we set the batch size to 32 during the training process, trained with 8*NVIDIA A100 GPU, and dynamically updated the learning rate using the cosine strategy during the training process. The baseline text encoder we use is based on the BERT model, which is natively used in CLIP and uses an MLP layer for feature mapping. We use (Rk, k=1,5,10) as our evaluation metrics of recall accuracy, they represent the percentage of correct retrieved items in top-k results. In the following experimental results, B2T and T2B refer to BEV-to-text retrieval and text-to-BEV retrieval, respectively.

### 4.3 EXPERIMENT RESULT

To summarise our outcome, we performed the BEV-retrieval task using BEV-CLIP. We adapted pre-trained BEVFormer to extract the BEV feature, and the concatenated results of fine-tuned parameters from Llama2+LoRA with the embedding generated from the knowledge graph were used as text features. We apply SCP to map the features generated from the two branches to generate a sequence of BEV features and text features with the same dimentionalities. We supervised the training jointly by caption generation loss BEV-Text contrastive loss based on cosine similarity.

| Method | LoRA | SCP | KG | CG | B2T_R1 | B2T_R5 | B2T_R10 | T2B_R1 | T2B_R5 | T2B_R10 |
|---|---|---|---|---|---|---|---|---|---|---|
| BERT* | – | – | – | – | 0.6409 | 0.9129 | 0.9557 | 0.5594 | 0.8915 | 0.9384 |
| Llama2* | ✓ | – | – | – | 0.7875 | 0.9757 | 0.9909 | 0.8194 | 0.9812 | 0.9906 |
| | ✓ | ✓ | – | – | 0.8059 | 0.9783 | 0.9947 | 0.8584 | 0.9909 | 0.9959 |
| | ✓ | ✓ | ✓ | – | **0.8599** | 0.9947 | 0.9994 | 0.8757 | 0.9968 | 0.9994 |
| | ✓ | ✓ | ✓ | ✓ | 0.8578 | **0.9954** | **0.9994** | **0.8766** | **0.9971** | **0.9997** |

Table 2: Comparison of all results. * denotes that the model parameters are frozen, SCP refers to shared cross-modal prompt, KG refers to Distmult knowledge graph embeddings, and CG refers to caption generation head.

All results of our comparative experiments are demonstrated in table 2. We observe our best result on the combination of Llama2, LoRA, SCP, distmult knowledge graph embedding and caption generation head, which are the accuracy proportions of 85.78% and 87.66% on BEV-to-text rank@1 and text-to-BEV rank@1, respectively. And we have exceeded the accuracy 99% of the remaining indicators, which out-performs the compared baseline method. These experimental results demonstrate that our proposed BEV-CLIP method can effectively solve the BEV retrieval problem.

## 4.4 ABLATION STUDY

In this section, we verify the effect of each of our proposed methods on the retrieval results and validate the effectiveness of the methods through multiple sets of ablation experiments. We discuss the effect of the large language model, knowledge graph, Shared Cross-Modal Prompt, and caption generation tasks respectively. For the baseline of the experiments, we employ the CLIP native text branch as our adaptive method and replace the original visual branch with BEVFormer encoder. Additionally, we insert a layer of Multilayer Perceptron (MLP) between the BEV branch and contrastive loss in order to align the feature size.

| Method | MLP | LoRA | B2T_R1 | B2T_R5 | B2T_R10 | T2B_R1 | T2B_R5 | T2B_R10 |
|---|---|---|---|---|---|---|---|---|
| BERT* | – | – | 0.6409 | 0.9129 | 0.9557 | 0.5594 | 0.8915 | 0.9384 |
| Llama2* | ✓ | – | 0.7244 | 0.9472 | 0.9713 | 0.7030 | 0.9507 | 0.9730 |
| | ✓ | ✓ | **0.7875** | **0.9757** | **0.9909** | **0.8194** | **0.9812** | **0.9906** |

Table 3: Ablation study results when using different text encoder. * denotes that the model parameters are frozen.

**Language Models** We adapt Llama2 as a large language model text encoder to compare with the BERT-based CLIP native text encoder. Additionally, we also utilize LoRA (Hu et al., 2021) to perform fine-tuning for language model. The experimental results are shown in table 3. We observed that the output results using Llama2 decoder have a significant performance improvement in all metrics compared to the CLIP text branch using BERT. Meanwhile, we also tried some methods for fine-tuning of large language models to enhance the encoding ability of Llama2 for scene description, such as LoRA. LoRA is to add a low-rank weight matrix as a learnable parameter in addition to the backbone network of LLM to realize the fine-tuning of LLM. Comparing Llama2 with and without LoRA fine-tuning, it can be found that fine-tuning Llama2 using LoRA also has significant gains. Further improvements of about 6% and 10% are achieved in the B2T_R1 and T2B_R1 metrics, respectively . One possible cause for this improvement is that the pre-training task of Llama2 contains fewer autonomous driving scenarios, and the fine-tuning on the autonomous driving dataset using the LoRA approach is able to bridge this gap better.

| Method | KGE | B2T_R1 | B2T_R5 | B2T_R10 | T2B_R1 | T2B_R5 | T2B_R10 |
|---|---|---|---|---|---|---|---|
| Llama2* + LoRA | – | 0.7875 | 0.9757 | 0.9909 | 0.8194 | 0.9812 | 0.9906 |
| | TransE | 0.8009 | 0.9804 | 0.9936 | 0.8455 | 0.9892 | 0.9965 |
| | Distmult | **0.8059** | **0.9783** | 0.9947 | **0.8584** | **0.9909** | 0.9959 |
| | ConvE | 0.8050 | 0.9780 | **0.9956** | 0.8473 | 0.9889 | **0.9968** |

Table 4: Ablation study results when using different knowledge graph. * denotes that the model parameters are frozen. KGE refers to knowledge graph embedding

**Knowledge graph embeddings** In order to verify the effect of adding knowledge graphs from autonomous driving related datasets to the text branch, we tried to add knowledge graphs derived from several different methods to the text branch for comparison experiments, including transE, distmult, and convE. we used the text branch of LoRA fine-tuned Llama2 as the baseline, and the experimental results are shown in table 4. We note that after adding the text embedding output from the 3 different maps to the text branch, there is a significant improvement compared to the baseline, especially in the T2B_R1, which is improved by 3% to 4% on average. Among these 3 kinds of knowledge graph, distmult achieves the optimal result.

| Method | LoRA | MLP | SCP | Distmult | B2T_R1 | B2T_R5 | B2T_R10 | T2B_R1 | T2B_R5 | T2B_R10 |
|---|---|---|---|---|---|---|---|---|---|---|
| | – | ✓ | – | – | 0.7244 | 0.9472 | 0.9713 | 0.7030 | 0.9507 | 0.9730 |
| | – | – | ✓ | – | **0.8291** | **0.9938** | **0.9997** | **0.8247** | **0.9959** | **1.0000** |
| Llama2* | ✓ | ✓ | – | – | 0.7875 | 0.9757 | 0.9909 | 0.8194 | 0.9812 | 0.9906 |
| | ✓ | – | ✓ | – | **0.8552** | **0.9944** | **0.9988** | **0.8751** | **0.9962** | **0.9991** |
| | ✓ | – | – | ✓ | 0.8059 | 0.9783 | 0.9947 | 0.8584 | 0.9909 | 0.9959 |
| | ✓ | – | ✓ | ✓ | **0.8599** | **0.9947** | **0.9994** | **0.8757** | **0.9968** | **0.9994** |

Table 5: Ablation study results of adding SCP when using multiple different baselines.* denotes that the model parameters are frozen, SCP refers to shared multi-modal prompt

**Shared cross-modal prompt** SCP is used as a strategy to align text features with BEV features in our method, and we verify its effectiveness in table 5. Comparing the three sets of ablation experiments of SCP under different conditions, it can be found that the direct introduction of SCP without modifying the encoder of the two branches is able to significantly improve the model metrics, and comparing the fine-tune strategy with the introduction of LoRA and the baseline model with knowledge graph embedding also both indicate that SCP brings effective improvement.

| Method | CG | Distmult | B2T_R1 | B2T_R5 | B2T_R10 | T2B_R1 | T2B_R5 | T2B_R10 |
|---|---|---|---|---|---|---|---|---|
| | – | – | 0.8552 | **0.9944** | 0.9988 | 0.8731 | 0.9962 | 0.9991 |
| | ✓ | – | **0.8573** | 0.9930 | **0.9988** | **0.8751** | **0.9974** | **0.9991** |
| Llama baseline | – | ✓ | **0.8599** | 0.9947 | 0.9994 | 0.8757 | 0.9968 | 0.9994 |
| | ✓ | ✓ | 0.8578 | **0.9954** | **0.9994** | **0.8766** | **0.9971** | **0.9997** |

Table 6: Ablation study results of adding caption generation as an auxiliary task. CG refers to the model trained with cation generation head. Llama baseline includes : Llama2* + LoRA + SCP.

**Caption generation** We add a caption generation head for assisted supervised training to fine-tune text branch for better semantic representation and to improve the feature alignment capability of the SCP The experimental results are shown in table 6. We observe that in the majority of cases, the adoption of the caption generation task leads to a better performance. In particular, we achieve the best result among all experiments for the T2B_R1.

## 5 CONCLUSION

In this paper, we propose a method for cross-modal retrieval with BEV features and text features for the first time. Specifically, on the BEV-branch, we proposed that the existing BEV model can be used to obtain BEV features without fine-tuning. On the text-branch, we proposed to use the pre-trained decoder-only LLM as the text encoder and concatenate the embedding generated by the knowledge graph training in the field of autonomous driving to form more robust text features. In addition, we propose SCP to fuse and align two modal information, and add caption heads to achieve multi-task training. We quantitatively verified the effectiveness of the method through a large number of ablation experiments on NuScenes dataset, and the analysis of the visualization results shows that the BEV retrieval task can deal with complex scenes in autonomous driving that cannot be solved by relying on single-frame and single-view images.

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

# A  APPENDIX

Here we demonstrate some typical cases retrieved by BEV-CLIP. We use the hand-labelled description text from NuScene dataset, with utilizing our text complement strategy to increase the accuracy and diversity of all description texts. It can be seen that the distributions and categories of the obstacles from the 6-view images basically match the description. However, we admit that a comprehensive multi-view camera dataset including complex road scene description does not exist by far. This makes us unable to present cases that are strongly related to actual usage scenarios.

| Description | 6-view camera |
|---|---|
| Moving cars, parked cars, heavy traffic, many cars, many trucks, several pedestrians, many traffic cones |  |
| Overtake heavy truck, stopped motorcycle, parking lot, peds, arrive at intersection, many cars, many trucks, many trailers, several buses, one motorcycle, several traffic cones |  |
| Very dense traffic, congestion, overtake car, peds, many cars, many trucks, several buses, one motorcycle, many pedestrians |  |
| Arrive at intersection, ped loading car trunk, ped with dog crossing crosswalk, bus, many cars, several trucks, several buses, many pedestrians |  |
| Wait at intersection, nature, scooters, peds on sidewalk, ped with umbrella, parked bicycle, taxi, jaywalker, delivery scooter, turn right, many cars, one truck, one bicycle, many motorcycles, many pedestrians |  |
| Ped crossing, crane crossing intersection, two big trucks moving, moving construction vehicle, many cars, many trucks, several trailers, many pedestrians |  |

Table 7: Rank 1 cases retrieved by BEV-CLIP on NuScene Dataset (with caption complement)

