# OpenReview forum: "BEV-CLIP: Multi-modal BEV Retrieval Methodology for Complex Scene in Autonomous Driving"
_ICLR.cc/2024/Conference — ICLR 2024 Conference Withdrawn Submission_

### Official Review · Reviewer_PBGj · 2023-10-30

**Soundness:** 2 fair
**Presentation:** 3 good
**Contribution:** 2 fair
**Rating:** 5
**Confidence:** 4

**Summary:**

In this paper, the authors propose a multi-modal BEV retrieval approach that utilizes descriptive text as an input to retrieve corresponding scenes, named BEV-CLIP. This approach leverages the semantic feature extraction capabilities of a large language model (LLM) to enable zero-shot retrieval, and integrates knowledge graphs to enrich and diversify the language embedding semantically. Experiments demonstrated the effectiveness of the approach.

**Strengths:**

1.	The authors claim that this is the first BEV retrieval method for autonomous driving.
2.	The authors leverage LLM and knowledge graph to achieve contrastive learning between text description and BEV feature to enable zero-shot retrieval.
3.	The authors build a validation pipeline and demonstrate the effectiveness of the proposed approach.

**Weaknesses:**

1.	The paper could benefit from stronger evidence, possibly in the form of quantitative experiments, to effectively illustrate the superiority of BEV features in comparison to 2D images for retrieval purposes..
2.	Contrastive learning has been widely explored to various tasks in autonomous driving, such as more meaningful driving decisions.
3.	The validation pipeline lacks persuasiveness, primarily due to its limited scope in which it solely evaluates and discusses a select few state-of-the-art methodologies.
4.	The outlook for scene retrieval in the context is currently uncertain, raising doubts about its viability as a worthwhile investment for autonoumous driving.
5.	The manuscript contains grammar errors and typos such as  "multimodal BEV retrieval methodology that utilize" should be "multimodal BEV retrieval methodology that utilizes" in the abstract.

**Questions:**

My main concerns and questions lie in the weaknesses. The author should discuss them in detail.

---

### Official Review · Reviewer_Z1vR · 2023-10-30

**Soundness:** 2 fair
**Presentation:** 2 fair
**Contribution:** 2 fair
**Rating:** 3
**Confidence:** 5

**Summary:**

In this paper, the authors focus on the retrieval of complex scenes using text inputs. They identify limitations in existing approaches, particularly the lack of a global feature representation and sub-par text retrieval ability. To address these limitations, the authors propose BEV-CLIP, which is the first multimodal BEV retrieval methodology that utilizes descriptive text to retrieve corresponding scenes. The key idea behind BEV-CLIP is to leverage the semantic feature extraction abilities of a large language model (LLM) to enable zero-shot retrieval of extensive text descriptions. Additionally, the method incorporates semi-structured information from a knowledge graph. The authors evaluate the performance of BEV-CLIP on the NuScenes dataset and demonstrate an accuracy of 87.66% in text-to-BEV feature retrieval.

**Strengths:**

1. Importance of retrieval for complex scenes in autonomous driving: The reviewers believe that this direction is critical for the field of autonomous driving.
2. To the best of the reviewer's knowledge, the field has not explored the use of Battery Electric Vehicle (BEV) representations extracted from images collected from multiple cameras for text retrieval purposes.

**Weaknesses:**

1. The primary motivation of the paper is to address the limitations of two-dimensional image features in effectively representing global features within autonomous driving scenarios. The authors propose the use of BEV representations. The reviewer wonders why the authors did not explore 3D occupancy as a potential solution to this issue, as suggested by Huang et al. in their paper "Tri-Perspective View for Vision-Based 3D Semantic Occupancy Prediction" (CVPR 2023). Therefore, while the reviewer acknowledges that this work is the first BEV retrieval method in the field of autonomous driving, they find the motivation unclear and insufficiently justified.
2. The second motivation of the paper is to explore methodologies that could enhance the currently unsatisfactory efficacy of text representations in the field of autonomous driving. The authors propose exploring CLIP to address the unsatisfactory efficacy of text representations. The reviewer wonders if the authors have considered exploring the potential of language parsing, as suggested by Liu et al. in their paper "Temporal Modular Networks for Retrieving Complex Compositional Activities in Video" (ECCV 2018) and Lin et al. in their paper "Visual Semantic Search: Retrieving Videos via Complex Textual Queries" (CVPR 2014). Without discussing and comparing these approaches, the reviewer is not convinced of the motivation behind this work.
3. Figure 1 is misleading. The effectiveness of the "BEV retrieval" setting is attributed to multiple perspectives rather than the BEV representation itself. The reviewer agrees that the use of multiple views can facilitate text retrieval, but the use of BEV representation remains questionable.

**Questions:**

The reviewer has identified three major concerns in the Weakness section and would like to know the authors' thoughts on these points. In summary, these concerns pertain to the motivation of the paper and the lack of literature. Please address each concern during the rebuttal stage. The reviewer will respond accordingly in the discussion phase.

---

### Official Review · Reviewer_cPYi · 2023-10-31

**Soundness:** 1 poor
**Presentation:** 2 fair
**Contribution:** 1 poor
**Rating:** 3
**Confidence:** 4

**Summary:**

The authors introduce a new task of multi-modal BEV retrieval methodology for complex scenes for the application of autonomous driving. They try to build a new retrieval dataset based on nuScenes.

**Strengths:**

The task of multi-modal BEV retrieval methodology for autonomous driving applications is quite new and interesting for the 3D community. The authors curate a new dataset and a network to tackle the problem.

**Weaknesses:**

1. The new task is not well-motivated. The authors claims that the proposed method could help identify complex corner-case/long-tail scenes, however, I don't see any qualitative results to support the claim. For example, it would be great if the authors could manually point out what the corner cases are for examples shown in Table 7 in Appendix.

2. One of the contribution is 'zero-shot retrieval using long text descriptions'. I don't see any descriptions to support the claim.

3. The manscript is not easy to understand. For example, the loss function is not clear to me. How positive and negative samples for contrastive loss are generated?

**Questions:**

Please refer to the weakness part. I admit that the authors propose a new dataset and a new baseline network to tackle the problem. However, I have concerns about the motivation and experiments (specified in the weakness part).

---

### Official Review · Reviewer_gTzc · 2023-11-05

**Soundness:** 2 fair
**Presentation:** 3 good
**Contribution:** 2 fair
**Rating:** 5
**Confidence:** 4

**Summary:**

The study introduces BEVCLIP, a pioneering multimodal BEV retrieval technique that leverages descriptive text to extract relevant scenes. This approach harnesses the semantic feature extraction capacities of a large language model (LLM) to enable zero-shot retrieval based on comprehensive text descriptions. Notably, the method integrates semi-structured data from a knowledge graph, thereby augmenting the semantic depth and diversity of the language embedding. The efficacy of this model is assessed using the NuScenes dataset.

**Strengths:**

1. The proposed BEV retrieval task is a distinct advancement. To my current awareness, it sets a pioneering standard as the first BEV retrieval method dedicated to the domain of autonomous driving.

2. The in-depth ablation studies—evaluating the impact of the large language model, knowledge graph, Shared Cross-Modal Prompt, and caption generation tasks—stand out for their thoroughness and analytical precision.

3. The composition of the manuscript is both well-organized and lucid, making it accessible and straightforward for readers.

4. With the model attaining a rank-1 result of 87.66% on the NuScenes dataset, it underscores its formidable potential for real-world implementation and practical relevance in the field.

**Weaknesses:**

1. The significance of the BEV retrieval task remains somewhat under-elaborated. Could the authors elucidate further on the task's implications and relevance for the autonomous driving sector?

2. The primary deviation from conventional image retrieval appears to be the amalgamation of multiple surrounding images for the retrieval process. A deeper exploration into the distinguishing features would be beneficial for readers.

3. The benchmarking exclusively hinges on the NuScenes dataset. Considering the variety of datasets available for autonomous driving, such as the Waymo Open Dataset and the KITTI Dataset, why were these omitted? A singular dataset benchmark might not offer a comprehensive evaluation.

4. The methodological design seems to lack specificity for BEV retrieval. Is the primary modification merely substituting the image encoder with a BEV encoder? If so, further clarification on its uniqueness is necessary.

5. The innovative aspects of the proposed modules warrant more distinction. For instance, the Shared Cross-modal Prompt draws parallels with [1] in its approach to group text and visual features using shared learnable centers. Similarly, the incorporation of knowledge graphs in cross-modal retrieval isn't a novel concept. The Caption Generation Head has also been studied in previous work like Coca[2] and BLIP2 [3]. A more in-depth comparison would be insightful.

Reference:
[1] T2VLAD: Global-Local Sequence Alignment for Text-Video Retrieval.
[2] CoCa: Contrastive Captioners are Image-Text Foundation Models
[3] BLIP-2: Bootstrapping Language-Image Pre-training with Frozen Image Encoders and Large Language Models

**Questions:**

My primary concerns revolve around the significance of the task at hand and the innovative nature of the proposed technique. I eagerly anticipate the authors' clarification on these aspects. Should these concerns be satisfactorily addressed, I would raise my rating.